# Direct allele introgression into pure chicken breeds using Sire Dam Surrogate (SDS) mating

Maeve Ballantyne[1,2], Mark Woodcock [2], Dadakhalandar Doddamani[2], Tuanjun Hu[1,2], Lorna Taylor[2], Rachel J. Hawken[3] & Mike J. McGrew [1,2✉]

Poultry is the most abundant livestock species with over 60 billion chickens raised globally per year. The majority of chicken are produced from commercial flocks, however many indigenous chicken breeds play an important role in rural economies as they are well adapted to local environmental and scavenging conditions. The ability to make precise genetic changes in chicken will permit the validation of genetic variants responsible for climate adaptation and disease resilience, and the transfer of beneficial alleles between breeds. Here, we generate a novel inducibly sterile surrogate host chicken. Introducing donor genome edited primordial germ cells into the sterile male and female host embryos produces adult chicken carrying only exogenous germ cells. Subsequent direct mating of the surrogate hosts, Sire Dam Surrogate (SDS) mating, recreates the donor chicken breed carrying the edited allele in a single generation. We demonstrate the introgression and validation of two feather trait alleles, Dominant white and Frizzle into two pure chicken breeds using the SDS surrogate hosts.

[1] Centre for Tropical Livestock Genetics and Health (CTLGH), Edinburgh, UK. [2] The Roslin Institute and Royal (Dick) School of Veterinary Studies University of Edinburgh, Easter Bush Campus, Midlothian, UK. [3] Cobb-Europe, Old Ipswich Road, Colchester, UK. ✉email: mike.mcgrew@roslin.ed.ac.uk

Poultry is the most abundant livestock species with over 60 billion chickens or 8 chickens raised for each human on the planet each year[1]. Commercial hybrid lines, which have been developed over many decades of selective breeding for meat or egg production, are most common and are extremely productive under controlled dietary and environmental conditions. Local populations of chicken that are genotypically, phenotypically and geographically distinct from each other hold nutritional, economic and cultural importance for smallholder farmers. Although their productivity is much lower than commercial lines, they are posited to be adapted to local climatic and pathogenic insults[2,3].

Globally, almost 1600 distinct regional breeds of chicken are recognised[4], and the utilisation of this diversity through the identification and validation of genetic variants for increased resistance to disease and heat stress will benefit both commercial and smallholder farmers[5]. The ability to efficiently transfer existing and introduce novel beneficial alleles into both commercial and traditional chicken breeds will aid ongoing efforts to validate disease, production and climatic adaptation traits in chicken for sustainable food production (reviewed in ref. [6]).

Primordial germ cells (PGCs) are the lineage-restricted stem cell population for the gametes (sperm and eggs) of the adult animal. The chicken is one of the few species in which the PGCs can be isolated from the developing embryo and then propagated in vitro[7,8]. In vitro propagated PGCs can be used for the editing of the chicken genome and the cryopreservation of chicken breeds[9–13]. As the chicken develops in a laid egg, PGCs can easily be microinjected into surrogate host chicken embryos, which are subsequently hatched, raised and bred to produce offspring in which one-half of their chromosomes derive from the donor PGCs. A limitation of this strategy is that the gonads of host embryos contain both introduced donor PGCs as well as the endogenous host PGCs, reducing the chance that offspring from subsequent matings will be derived from donor PGCs. To improve the transmission of donor genetic material, it is advantageous to reduce or eliminate the host's own PGCs. Sterile male hosts have been produced in pig, bovine and goat species[14,15]. In chicken, a sterile female surrogate host, a dam line containing a knockout of the DDX4 gene, was recently shown to lay eggs solely deriving from introduced donor female PGCs isolated from a rare breed of chicken[16,17]. Developing sterile male and female (sire and dam) surrogate hosts would permit the direct reconstitution of a pure chicken breed from exogenous PGCs through the mating of surrogate hosts and the simultaneous generation of offspring homozygous for any genomic variant introduced into the PGCs through genome editing.

## Results

**iCaspase9 expressed from the DAZL locus selectively ablates the germ cell lineage in birds.** We aimed to produce a surrogate host chicken line in which the germ cell lineage of both males and females could be conditionally ablated. The inducible caspase-9 (iCaspase9) protein consists of a truncated human caspase-9 protein fused to the FK506-binding protein (FKBP) drug-dependent dimerisation domain[18]. The chemical compound, AP20187 (B/B), induces the dimerisation of FKBP and subsequent activation of the adjoining caspase-9 protein leading to induced apoptotic cell death. iCaspase9 has previously been used as a cellular suicide gene for human stem cell therapy[18–20] and to ablate cell lineages in transgenic animals[21,22]. We also produced a transgene containing the equivalent region of the chicken caspase-9 protein, aviCaspase9 (Supplementary Materials and Methods). To specifically drive the expression of iCaspase9 gene in the germ cell lineage, we used CRISPR/Cas9-mediated

homology-directed repair (HDR) to target the iCaspase9 construct to the 3′-end of the last coding exon of the DAZL gene in in vitro propagated chicken PGCs (Fig. 1a). The chicken DAZL gene is a putative avian germ cell determinant and is highly expressed in migratory PGCs and germ cells in the embryonic gonad[23,24]. The iCaspase9 transgene was preceded by a 2A peptide sequence and followed by a second 2A peptide and a GFP reporter gene to mark cellular expression. Female PGCs were transfected with the constructs, and GFP-expressing PGCs were purified by flow cytometry 4 weeks post transfection. Targeted GFP+ PGCs were assayed for proliferation after addition of the B/B dimerisation compound. The growth of PGCs containing either the iCaspase9 or aviCaspase9 transgene was inhibited at nanomolar concentrations of the B/B compound compared with control PGCs targeted with a GFP reporter alone (Fig. 1b).

Female PGCs were sorted for GFP fluorescence and subsequently injected into the embryonic vascular system of stage 16+ HH (embryonic day 2.5)-recipient embryos of the DDX4 knockout chicken line. Female chicken ablated for DDX4 contain no oocytes post hatch and will produce offspring solely deriving from introduced female PGCs[17]. The injected embryos were sealed and incubated to hatch, then raised to sexual maturity. The wild-type and DDX4 female surrogate hosts were bred to wild-type cockerels, and the $G_1$ offspring were screened for the presence of the iCaspase9 transgene. We observed no transgenic offspring from the wild-type host females indicating that the genetically modified PGCs could not compete with the endogenous PGCs. In contrast, 11% of the $G_1$ chicks contained the transgene ($n = 8$ of 74) (Supplementary Tables 1 and 2 and Supplementary Fig. 1). The iCaspase9 and aviCaspase9 $G_1$ chicks were raised to sexual maturity and the $G_1$ cockerels were mated to wild-type hens and $G_2$ embryos were examined for GFP expression. GFP fluorescence was observed only in the gonads of developing male and female embryos which PCR confirmed were positive for the transgene. GFP+ cells were also positive for the germ cell markers SSEA1 and DDX4, indicating the transgene was specifically expressed in the germ cell lineage and gonad of the embryo (Fig. 1c, d and Supplementary Figs. 2, 3, 4, 6). Different concentrations of B/B dimerisation drug were delivered into chicken embryos in ovo which were incubated and examined for both GFP expression and germ cell ablation. We observed that the iCaspase9 transgene was more highly expressed in the gonadal germ cells (Supplementary Fig. 4) and more effective for in ovo germ cell ablation than the aviCaspase9 transgene (Fig. 1d, e and Supplementary Figs. 5–7). Ectopic cell death was not observed in either iCaspase9 control or B/B-treated embryos several days post-treatment (Supplementary Fig. 8). iCaspase9 hens also laid eggs at a rate comparable to control age-matched hens indicating normal oogenesis was not affected by the transgene (Supplementary Table 3).

**Removal of the dominant white (DOW) mutation from a white-feathered chicken breed.** The plumage colour of local indigenous chicken populations shows great regional diversity and smallholder farmers value coloured plumage for aesthetic, cultural and economic reasons[25,26]. The dominant white (DOW) feather trait is fixed in the most common commercial layer chicken breed, the White Leghorn (WL), giving them their characteristic pure white plumage[27]. The DOW allele (I) has been identified as a putative 9 bp insertion within exon 10 of the PMEL17 gene, which creates a three amino acid (WAP) insertion in the transmembrane region of the PMEL17 protein[28]. PMEL17 is a melanocyte transmembrane glycoprotein involved in eumelanin deposition and fibril formation in melanosomes[29]. In chicken, the plumage of homozygote DOW WL chicken is

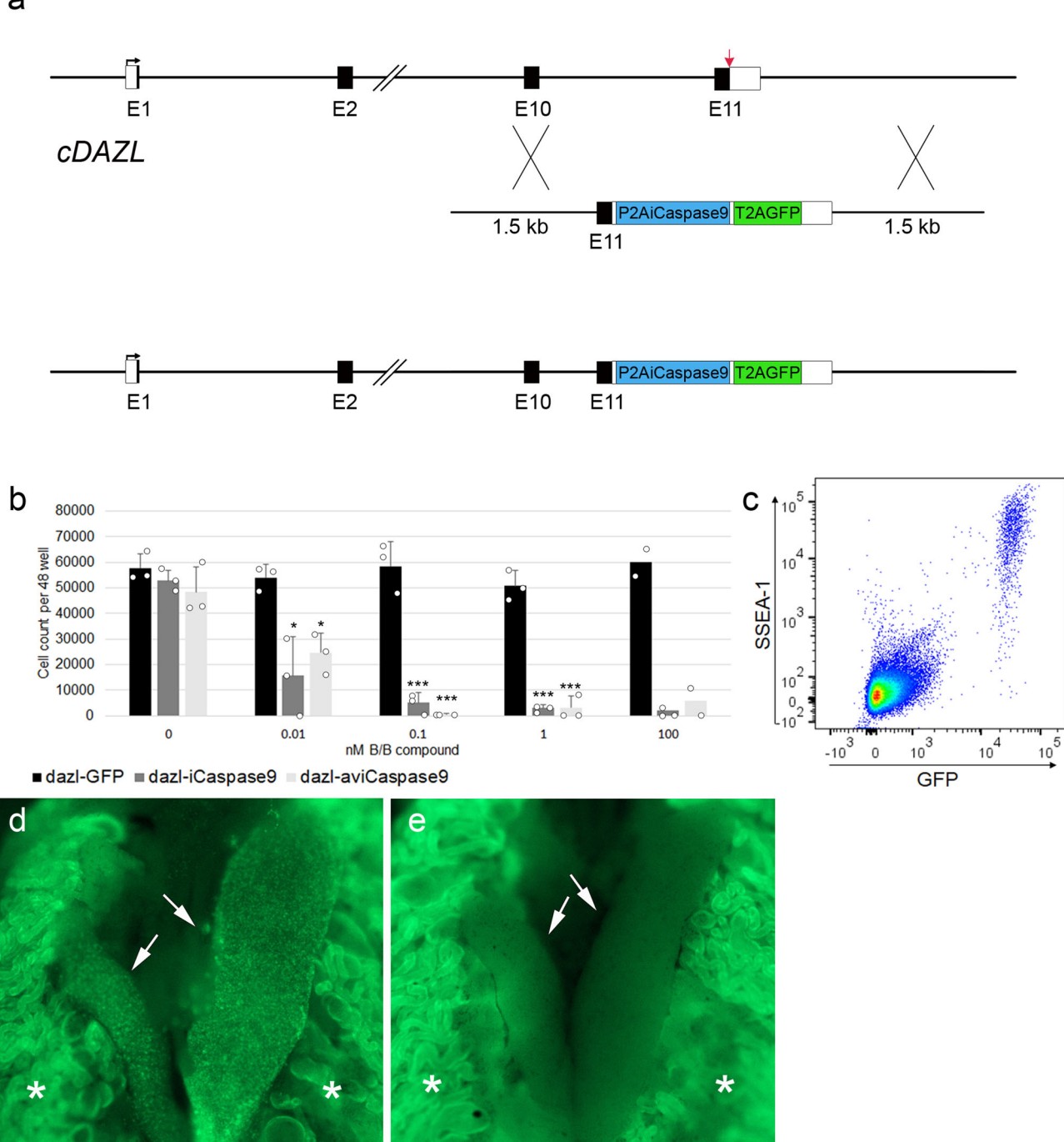

**Fig. 1 iCaspase9-mediated ablation of the germ cell lineage. a** CRISPR/Cas9-mediated recombination of an iCaspase9 and GFP reporter gene into the final coding exon of the *DAZL* locus. Red arrow indicates the guide target. **b** 500 PGCs targeted with a human iCaspase9 or a chicken iCaspase9 (aviCaspase9) transgene were cultured in the presence of different concentrations of the B/B dimerisation compound for 10 days. Data are presented as the mean ± SD, $n = 3$ ($n = 2$ for 100 nM). One-way ANOVA, *$P < 0.05$, ***$P < 0.001$ with respect to Dazl-GFP cells. **c** Day-10 embryonic gonads were examined for expression of GFP and SSEA1 immunofluorescence, $n = 4$. **d** Control and (**e**) B/B-treated iCaspase9 $G_2$ embryos imaged at day 10 of incubation. Arrows indicate the embryonic gonads. *, autofluorescence in the underlying mesonephros. $n = 5$ for each genotype. Scale bar, 100 μm.

completely white whilst heterozygote DOW birds contain a few scattered black spots in their feathers[30].

The removal of the DOW 9 bp insertion from a WL chicken genome is expected to restore the underlying feather plumage colour in homozygous birds. We chose to address this hypothesis using the inbred MHC congenic (Line 6) research line of WL chicken. Sequencing of WL Line 6 chickens revealed that this line is homozygous for the DOW allele and Sex-linked barring B2

allele, which generates a barred plumage in female chicken (W/B2) and white-feathered males (B2/B2) (Supplementary Fig. 9)[31]. Thus, removal of the DOW mutation from WL Line 6 chicken should, in principle, generate barred feathered females and white-feathered males.

To selectively remove the DOW allele, we propagated PGCs from a single male and female WL Line 6 stage $16^+$ HH (day 2.5) embryos[8]. A CRISPR guide along with a high fidelity Cas9

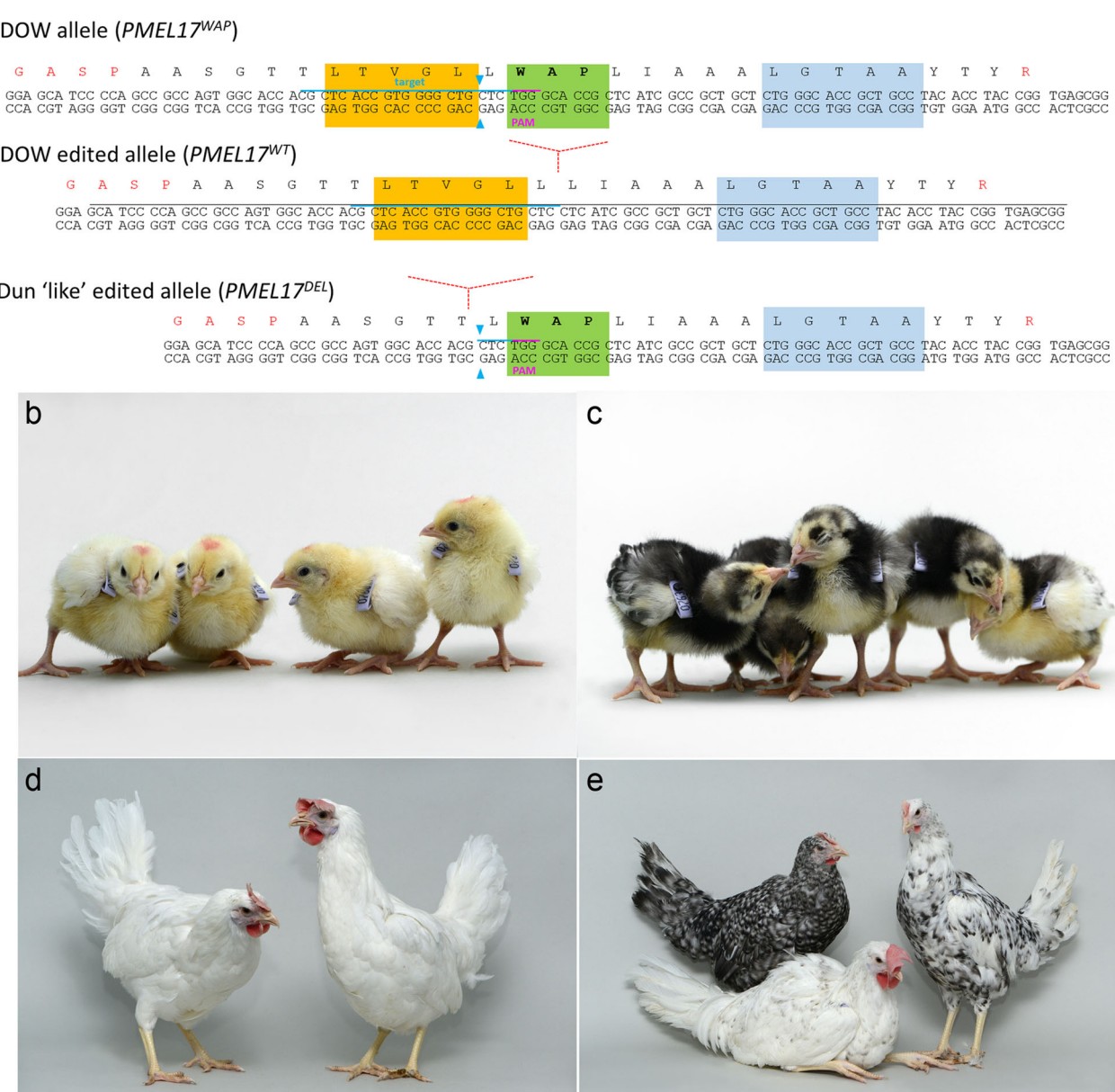

**Fig. 2 Genome edit to remove the DOW allele from a White leghorn chicken breed. a** *PMEL17* locus containing the DOW allele (WAP insertion) within the 10th exon which encodes for the PMEL17 transmembrane domain (amino acids in black). Genome-edited *PMEL17* locus removing the WAP (highlighted green). Genome-edited second allele creating a five amino acid Dun-like deletion (deleted amino acids highlighted yellow). The five amino acids deleted in the Dun allele are highlighted in blue. **b**, **d** Wild-type WL Line 6 chicks and adults. **c**, **e** *PMEL17* edited chicks and adults; females (standing) were barred and speckled feathers, males (sitting) were white with black dots. All photographs of live birds at The Roslin Institute.

(SpCas9-HF1) and a ssODN donor template containing the wild-type *PMEL17* sequence, were transfected into the PGC cultures. Distinct 95 bp ssODN donor templates were used for transfection of the female and male PGC cultures with the purpose of tracking the parental derived alleles in the $G_1$ generation. These templates differed by a single synonymous nucleotide change which was introduced into the female PGCs, whereas the wild-type *PMEL17* nucleotide sequence was introduced into the male PGCs (see Supplementary Table 4 for ssODN sequences). After clonal isolation and culture, an edited wild-type allele (*PMEL17^-WAP*) was identified in ~50% of PGCs clones in both male and female PGC lines. Unexpectedly, in most male and female clones the second *PMEL17* allele contained an identical and novel 15 bp deletion adjacent to the Cas9 cut site (Fig. 2a). This deletion removes the guide binding site and creates a five amino acid in-frame deletion in the

transmembrane domain whilst leaving the PAM sequence and the DOW insertion allele intact (*PMEL17^Del+WAP*). This deletion was identified to occur at this Cas9 cleavage site by microhomology-mediated repair predictors[32,33]. PCR analysis of five potential off-target sites in a *PMEL17^Del+WAP* edited clone and resulting offspring (see below) did not identify any novel off-target mutations (Supplementary Fig. 10).

We used these PGCs to generate a *PMEL17* allelic series containing the two edited changes. To directly produce genome-edited $G_1$ chicks, we mixed edited female and male PGCs and co-injected these with B/B dimerisation compound into the embryonic vascular system of stage $16^+$ HH iCaspase9 and aviCaspase9 chicken embryos in windowed eggs (Fig. 3). The host shells were sealed and incubated until hatching (Supplementary Table 1). The iCaspase9 surrogate host chicks were raised to

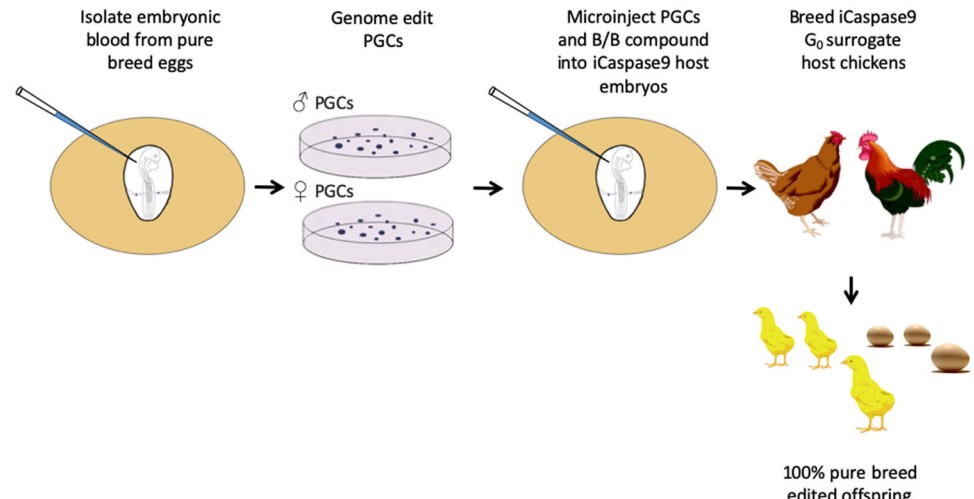

**Fig. 3 Schematic of SDS mating.** PGCs are isolated and propagated in vitro from a pure chicken breed. After genome editing and clonal isolation, the male and female PGCs are mixed with the B/B compound and injected into iCaspase9 surrogate host embryos. The embryos are hatched, raised to sexual maturity then mated. Laid eggs and hatched offspring are from the donor breed of interest and contain the desired genome edit. Drawing by MJM.

| iCaspase9 host mating groups | No. of eggs laid per week* | No. of eggs incubated | Fertility[a] (% eggs incubated) | No. of chicks hatched[b] (% fertile eggs) | [c]Transmission of host iCaspase9 transgene (%) |
|---|---|---|---|---|---|
| **Table 1 Germline transmission rates from surrogate hosts injected with donor *PMEL17* edited WL PGCs.** | | | | | |
| $G_0$1-15♂ x $G_0$2-13♀ $G_0$2-22♀ | 7.0 | 36 | 27 (75%) | 17 (63%) | 0 |
| $G_0$2-16♂ x $G_0$2-13♀ $G_0$2-22♀ | 5.78 | 66 | 51 (77%) | 30 (59%) | 0 |
| *aviCaspase9 mating group* $G_0$2-08♂ $G_0$2-09♂ x $G_0$1-19♀ $G_0$1-21♀ $G_0$1-22♀ | 4.43 | 82 | 68 (94%) | 56 (82%) | 9 (16%) |
| *G1 generation mating groups* $G_1$24♂ x $G_1$9♀ $G_1$12♀ | 5.13 | 43 | 29 (67%) | 27 (93%) | 0 |
| $G_1$24♂ x 2 Line 6 WL ♀ | 5.72 | 59 | 28 (47%) | 25 (89%) | 0 |

*Lay rate: eggs were counted over a 60 day period when hens were between 7 and 12 months of age and divided by the number of fertile hens present in pen. The maximum possible lay rate is 7.0 eggs per week.
[a]Fertility: no. of eggs with embryos at day 18 of incubation detected by candling.
[b]Hatchability: no. of chicks hatched from fertile day 18 eggs.
[c]No. of offspring PCR+ for iCaspase9 transgene.

sexual maturity and directly mated to each other in natural mating groups; a process we call Sire Dam Surrogate (SDS) mating. Three independent mating groups were produced and fertile eggs were incubated and hatched (Table 1). All hatchlings from the iCaspase9 mating groups displayed a black plumage pigmentation (Fig. 2b, c). In addition, a PCR analysis did not detect the iCaspase9 transgene in any of the offspring (Table 1). When mature, the cockerels displayed a floppy comb that is consistent for Line 6 WL birds (Fig. 2d, e). Furthermore, a principal component (PC) analysis indicated that all offspring from the iCaspase9 clustered with control WL Line 6 birds (Fig. 4). In contrast, the aviCaspase9 surrogate host group produced many chicks with yellow feathers and PCR analysis of the offspring detected the iCaspase9 transgene, indicating the endogenous germ cells were not completely ablated in this host (Table 1). The egg lay rate and fertility from natural matings for the iCaspase9 mating groups were high and similar to that

observed for the aviCaspase9 mating group. However, the hatchability from the iCaspase9 surrogate host mating groups was lower than expected (~60%) in comparison to the aviCaspase9 group (82%). This suggests that the in vitro culture/manipulations of the PGCs may have lowered overall hatchability.

The adult $G_1$ birds displayed three colour phenotypes; (Fig. 2e) cockerels were spotted white, and hens had barred or speckled feathers. Sanger sequencing analysis and restriction digest genotyping revealed that the barred/speckled females had the genotypes $PMEL17^{-WAP/-WAP}$, $PMEL17^{-WAP/Del+WAP}$, $PMEL17^{Del+WAP/Del+WAP}$ (Supplementary Fig. 11a–c). The $PMEL17^{-WAP/-WAP}$ birds all contained a single tracking nucleotide, indicating that these offspring are descended from one male and one female PGC (Supplementary Fig. 11a). Crossing a $PMEL17^{-WAP/-WAP}$ $G_1$ male and $PMEL17^{-WAP/Del+WAP}$ and $PMEL17^{-WAP/-WAP}$ females confirmed three feather colour

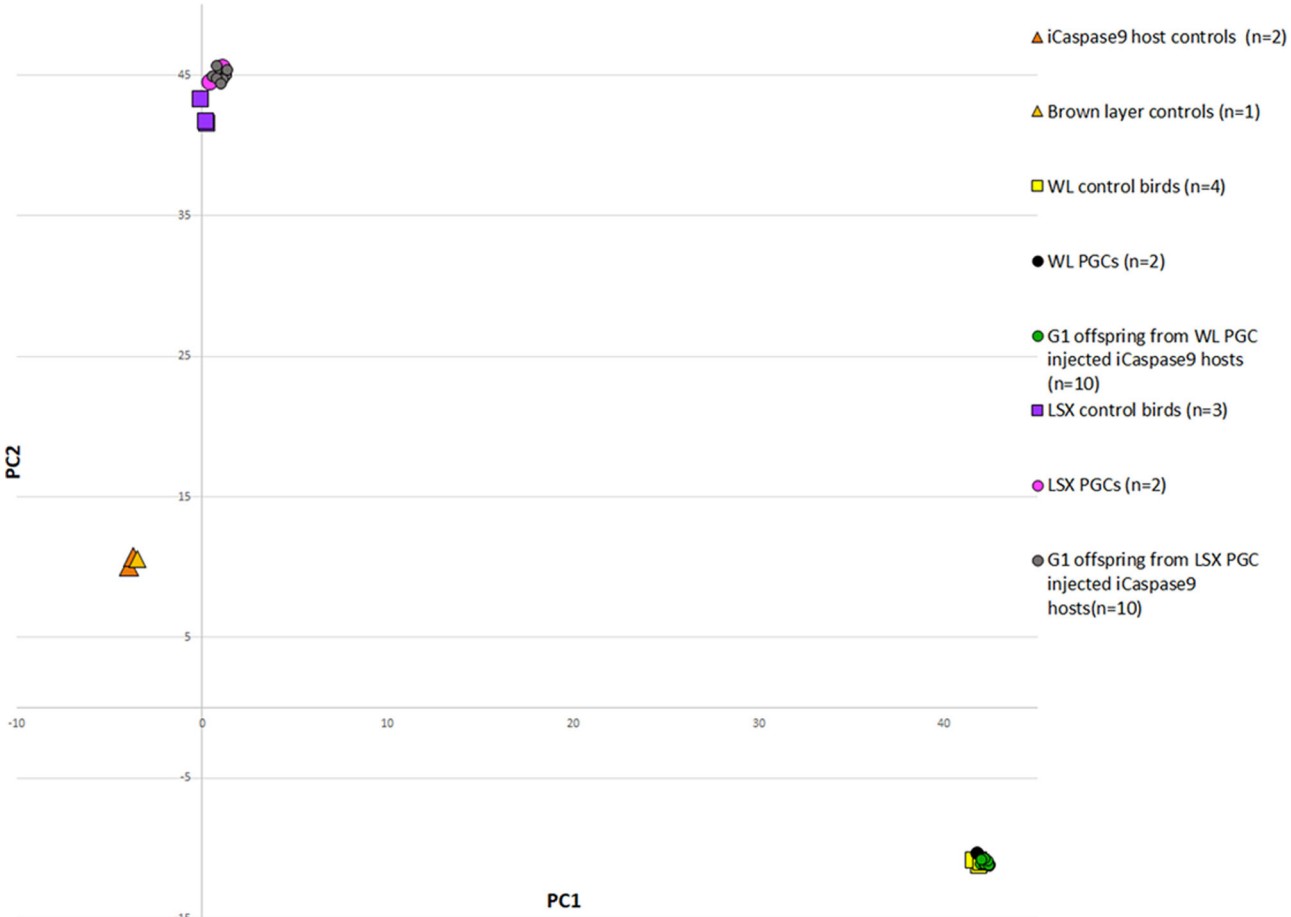

**Fig. 4 PC analysis of G1 offspring from iCaspase9 surrogate host birds.** DNA samples were genotyped on a 66 K SNP chip and analysed for PCs. Three chicken breeds were analysed: the brown layer breed which includes the iCaspase9 hosts, the LSX breed, and the WL Line 6 breed. Offspring from iCaspase9 hosts injected with LSX PGCs (grey) clustered with LSX control birds. Offspring from iCaspase9 surrogate hosts injected with WL Line 6 PGCs (green) clustered with WL Line 6 control birds.

morphs (cockerels, spotted white; hens, barred or speckled feathers) were present in the $G_2$ generation which corresponded with the expected feather colour phenotypes. Breeding $PMEL17^{-WAP/-WAP}$ $G_1$ males with Line 6 wild-type birds generated all white-feather offspring showing that the edited alleles were recessive to the DOW allele (Supplementary Fig. 12). This result suggests the length of the transmembrane domain is important for optimal function. The wild-type PMEL17 transmembrane domain is 25 amino acids (aa) in length, the WAP insertion increases this length to 28 aa and creates a dominant-negative protein. The naturally occurring *Dun* loss of function allele reduces the transmembrane length to 20 aa. Here, we restore the wild-type 25 aa length in the -WAP allele ($PMEL17^{-WAP}$) and create a novel mutation resulting in a 27 aa transmembrane domain, $PMEL17^{Del+WAP}$. Both of which restore eumelanin production as attested to by the dark barred feather female offspring.

**Introduction of the Frizzle (FRZ) feather trait into a traditional European breed.** Frizzled (FRZ) feather chicken is highly valued in Western Africa and this trait is posited to confer adaptability to tropical climates[34,35]. The FRZ feather allele (*F*) is a naturally occurring splice variant caused by an 84 bp deletion in the α-keratin 75 genes (*KRT75*) leading to a curved feather rachis and barbs[36]. Homozygote FRZ birds have severely frizzled and brittle feathers that are easily broken, however, heterozygote FRZ chicken display an attractive ruffled, frizzled feather phenotype.

An autosomal recessive modifier gene (mf) that lessens the effects of the FRZ allele is present in many European chicken breeds, leading to a minor frizzled phenotype in young chicken and a crumpling of the barbs of the anterior flight feathers[37,38]. To produce heterozygote FRZ chicken, we first isolated and propagated male and female PGCs from eggs deriving from a flock of Light Sussex (LSX) chicken. The LSX chicken is a traditional dual-purpose British breed with black tail feathers and black stripped hackles[39] (Fig. 5c). We again used high fidelity Cas9, SpCas9-HF1, and a 100 bp ssODN template to introduce the FRZ deletion into the *KRT75* gene of female LSX PGCs. After clonal isolation and propagation, we found that 10% of the PGCs clones were homozygous for the FRZ deletion (Fig. 5a). A homozygous *KRT75*-edited female clone selected for the generation of FRZ edited birds was screened for potential off-targeting mutations. No novel mutations were detected in the five off-target sites analysed (Supplementary Fig. 13).

To directly produce heterozygote FRZ edited LSX birds, we used the iCaspase9 surrogate host eggs and SDS mating. The homozygous *KRT75*-edited female LSX PGCs and wild-type male LSX PGCs were mixed and co-injected with the B/B compound into the embryonic vascular system of stage 16+ HH iCaspase9 chicken embryos in windowed eggs (see Fig. 3). The eggs were sealed and incubated to hatch (Supplementary Table 1). The iCaspase9 $G_0$ host birds were raised to sexual maturity and naturally mated. Fertile eggs from the SDS matings of two independent groups were incubated and hatched. Egg laying,

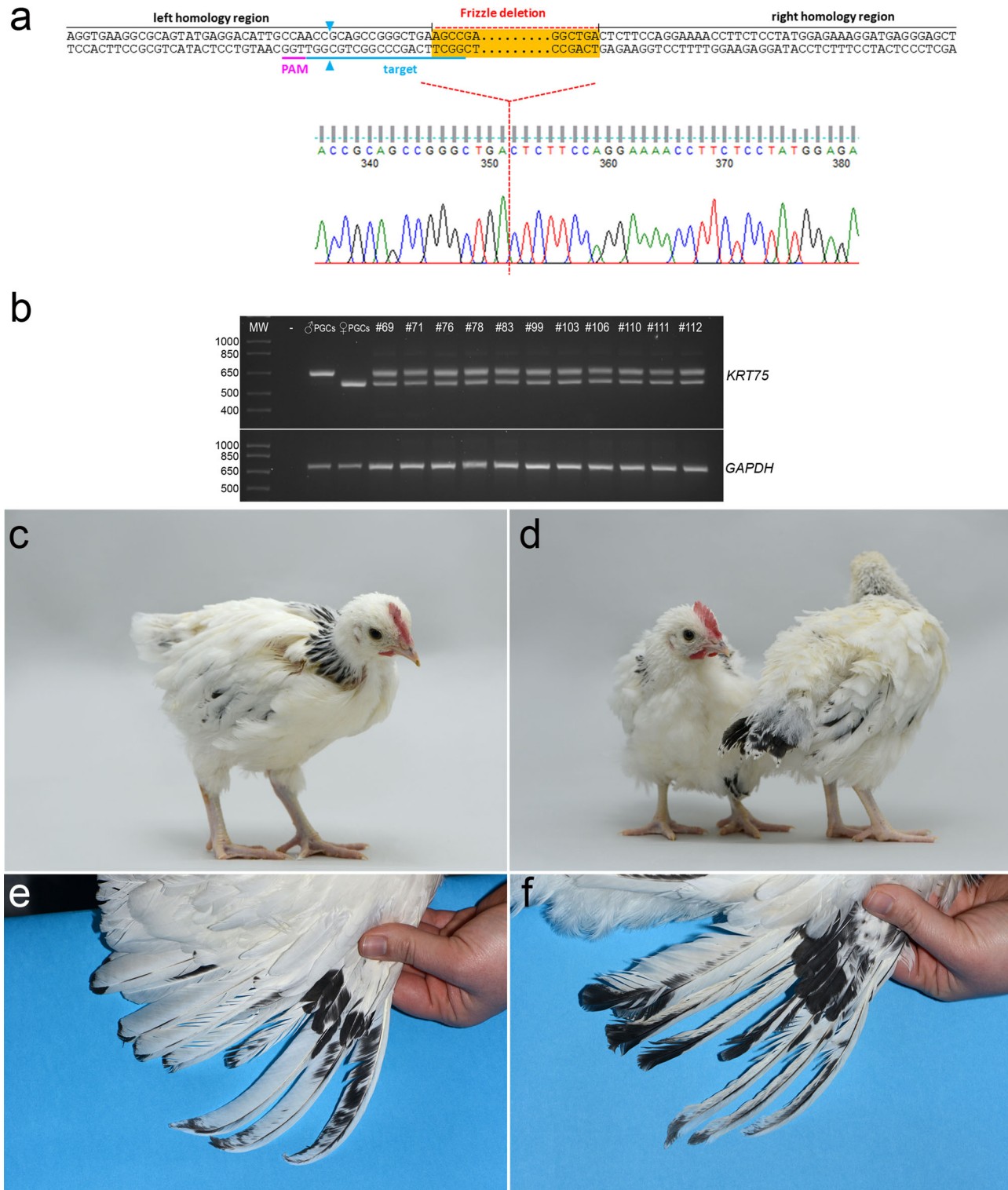

**Fig. 5 Genome edit to introduce the FRZ allele into the LSX chicken breed. a** *KRT75* locus surrounding the FRZ mutation. A single guide (blue) that targets cleavage 12 bp from the intended deletion was transfected along with a ssODN containing 50 bp homology on each side of the deletion. A sequencing chromatogram shows the bi-allelic deletion in female LSX PGCs. **b** PCR analysis of *KRT75*-edited LSX PGCs and G$_1$ offspring from iCaspase9 hosts. The wild-type locus produces a PCR product of 657 bp. The edited locus produced a PCR product of 573 bp, $n = 11$. **c**, **e** Control 6-week-old offspring and wing (18 weeks) from LSX birds. $n = 10/10$ female birds. **d**, **f** FRZ heterozygote LSX G1 offspring (6 weeks) and wing (18 weeks) displaying crumpled flight feathers, $n = 5$ of 5 female birds. All photographs of live birds at The Roslin Institute.

**Table 2 Germline transmission rates from surrogate hosts injected with donor *KRT75*-edited ♀LSX PGCs and wild-type ♂LSX PGCs.**

| iCaspase9 host mating groups | No. of eggs laid per week* | Eggs incubated | Fertility[a] (% of eggs incubated) | No. of chicks hatched[b] (% fertile eggs) | [c]Transmission of host iCaspase9 transgene (%) |
|---|---|---|---|---|---|
| $G_0$4-16♂ x $G_0$4-12♀ $G_0$4-19♀ | 6.65 | 116 | 64 (55%) | 43 (67%) | 0 |
| $G_0$5-15♂ x $G_0$5-5♀ $G_0$5-19♀ | 6.48 | 27 | 16 (59%) | 10 (63%) | 0 |

*Eggs in pen were counted over a 60 day period when hens were between 10 and 13 months of age and divided by the number of fertile hens present in pen. The maximum possible lay rate is 7.0 eggs per week.
[a]Fertility: no. of eggs with embryos at day 18 of incubation detected by candling.
[b]Hatchability: no. of chicks hatched from fertile day 18 eggs.
[c]No. of offspring PCR+ for iCaspase9 transgene.

fertility and hatchability were appropriate for layer chicken and are shown in Table 2. PCR analysis of both embryos and hatched chicks found that all offspring were heterozygous for the FRZ allele, indicating that they were derived from a single wild-type male PGC and a single *KRT75-* edited female PGC (Fig. 5b). A PC analysis showed that the iCaspase9 offspring clustered with control LSX birds indicating all offspring derived from donor LSX PGCs (Fig. 4). The hatched offspring displayed typical LSX colouration and a feather phenotype representative of a modifier FRZ heterozygote; obvious FRZ feather phenotype at 4 weeks of age and the flight feathers were crumpled as adults (Fig. 5c–f).

## Discussion

Our results demonstrate the usefulness of sterile surrogate hosts in a bird species, producing pure breed homozygous and heterozygous genome-edited offspring by the direct mating of sire and dam surrogates (Fig. 3). SDS mating is particularly amenable to species in which the embryo develops ex utero, such as birds, fish, amphibians and reptilian species[40–43]. As the chicken is a model organism for the study of embryogenesis, neural development and immunity in bird species, this technique offers a reliable platform for investigating gene function and disease resistance in poultry. Previously the production of homozygous genome-edited birds has required a minimum of two generations[44–49]. As the generation time for chicken is 5–6 months, SDS breeding greatly accelerates the production of homozygous offspring for the validation of genetic variants and vastly increases the number of offspring containing the homozygote genome edit. Using genetic modifications to achieve sterility adheres to the principles of the 3Rs (Replacement, Reduction, Refinement) as it reduces the number of animals required to generate the desired offspring and eliminates the use of chemotherapeutic reagents such as busulphan that are traditionally used to eliminate the endogenous germ cells.

The cryo-conservation of avian species is particularly challenging due to both the yolk filled egg and the difficulty in cryo-preserving avian sperm. SDS mating bypasses these challenges permitting the regeneration of avian breeds directly from frozen reproductive material. Indeed, here we demonstrated the regeneration of two pure chicken breeds from cryopreserved PGCs: the White Leghorn and the Light Sussex. We also show that breed regeneration can be coupled with the introduction and removal of genetic traits. As fertile chicken eggs can be shipped globally between avian breeding facilities, in the future an edited chicken breed could be generated directly from fertile eggs without first establishing a breeding population of birds. However, SDS matings produced large numbers of full siblings deriving from the same male and female PGC genotypes. This could be useful for

allelic validation as there would be reduced phenotypic variation between edited and non-edited offspring. However, chicken flocks are maintained as highly genetically diverse populations to avoid inbreeding which causes dramatic reductions in fertility and hatchability. It may prove necessary to multiplex multiple genotypes through single surrogate hosts in order to regenerate outbred chicken populations.

## Methods

**Chicken breeds and welfare.** The MHC Line 6 congenic line of WL birds[50] and the LSX dual-purpose chicken breed[39] are maintained as a closed breeding population of 150 birds in the NARF SPF facility (UK) and provided fertile eggs used for PGC derivations. Individual Line 6 birds were sequenced to confirm the presence of the DOW allele and the absence of the recessive white allele[51]. The barred feather allele was identified in Line 6 by analysis of the *CDKN2a* locus for SNP2 and SNP4[31]. The iCaspase9 lines of chickens were generated using Hy-line Brown layer PGCs. Heterozygous and homozygous cockerels carrying the iCaspase9 transgene were crossed to Hy-line hens to produce heterozygote iCaspase9 surrogate host eggs for injection and hatching. *DDX4* ZZ− heterozygote males were crossed to Hy-line hens to produce fertile host eggs for injection. All animal management, maintenance and embryo manipulations were carried out under UK Home Office license (70/8528) and regulations. Experimental protocols and studies were approved by the Roslin Institute Animal Welfare and Ethical Review Board Committee.

**Chicken PGC culture and transfection.** PGC lines were derived from the blood of stage 15-16+ HH embryos and propagated in culture, as previously described in ref. [8]. The embryos and PGC lines were sexed using a W-chromosome-specific PCR as previously described in[52]. Both male and female PGC cultures were derived from Hy-line, LSX and Inbred Line 6 WL embryos. Each PGC line was in culture for ~3 weeks before freezing in aliquots of 50,000 cells resuspended in 125 µl of Stem-Cellbanker (Amsbio) and stored at −150 °C.

For the generation of iCaspase9 birds; 1.0 µg of iCaspase9 or aviCaspase9 targeting vector and 1.0 µg of *DAZL* guide CRISPR/Cas9 vector were transfected into $1.5 \times 10^5$ Hy-line PGCs using Lipofectamine 2000 transfection reagent (Thermo Fisher Scientific). PGCs were cultured for 3 weeks, and GFP+ PGCs were purified by flow cytometry using a BD FACSAria III Cell Sorter gated for GFP fluorescence. For CRISPR/Cas9 editing of *KRT75* and *PMEL17*, $2 \times 10^5$ of LSX or Line 6 PGCs were transfected with 1.5 µg of CRISPR/Cas9 SpCas9-HF1 vector and 0.5 µg of ssODN donor template[10]. After 24 h in culture, the cells were treated with 0.6 µg/ml puromycin for 48 h to select for cells transfected with CRISPR vector. After selection, PGCs were sorted using the BD FACSAria III Cell Sorter into a 96-well plate at one cell per well After culturing for 3 weeks, clonal PGC populations were cryopreserved and used for genomic DNA isolation.

**Production of surrogate host chicks.** Targeted male and female PGC lines were thawed from storage at −150 °C and cultured for 5–10 days before injection into stage 15-16+ HH (day 2.5) surrogate host embryos in windowed eggs. In all, 1.0 µl of PGCs were directly injected into the dorsal aorta, and all shells were resealed with medical Leukosilk tape (BSN Medical) before incubating until hatch. For experiments using iCaspase9 surrogate host embryos, 1.0 µl of 25 mM B/B (in DMSO) (Takara Bio) was added to 50 µl PGC suspension (0.5 mM BB, 2% DMSO f.c.) before injection and kept at room temperature during the injection period. Subsequently, 50 µl of 300 µl Penicillin/Streptomycin (containing 30 µl of 0.5 mM B/B compound) was pipetted on top of the embryo. B/B compound was resuspended in DMSO instead of ethanol as ethanol proved toxic to the donor PGC suspension. Founder males and females were naturally mated in pens to produce

$G_1$ offspring. All offspring were screened by PCR for the presence of iCaspase9 transgene. See Supplementary Table 1 for details on individual experiments.

**Reporting summary**. Further information on research design is available in the Nature Research Reporting Summary linked to this article.

## Data availability

The data that support this study are available from the corresponding author upon reasonable request. The animal models in this study are available to licenced animal research facilities under MTA. Source data are provided with this paper.

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

## Acknowledgements

We thank the members of the Roslin chicken facility (A. Sherman, M. Hutchison, F. Brain, K. Hogan and F. Thomson) for care and breeding of the chickens, Jun Chen and Brenda Flack (Cobb-Vantress) for the SNP chip analysis, Norman Russell for photographing chicken, and Megan Davey, Sudeepta Panda and Guillermo Tellez for critiquing the paper.

We thank Donald Nkrumah for the guidance to edit the FRZ feather allele, Jeff Barrow for the DOW allele, and Appolinaire Djikeng, Bruce Whitelaw, Steve Kemp and the members of the Centre Management Group of the CTLGH for constructive comments on the project. This research was funded in part by the Bill & Melinda Gates Foundation and with UK aid from the UK Foreign, Commonwealth and Development Office (Grant Agreement OPP1127286) under the auspices of the Centre for Tropical Livestock Genetics and Health (CTLGH), established jointly by the University of Edinburgh, SRUC (Scotland's Rural College), and the International Livestock Research Institute. The findings and conclusions contained within are those of the authors and do not necessarily reflect positions or policies of the Bill & Melinda Gates Foundation nor the UK Government. This work was supported by the Institute Strategic Grant Funding from the BBSRC (BB/P0.13732/1 and BB/P013759/1) and Innovate UK Agri-Tech funding (BB/M011895/1).

## Author contributions

M.B., M.W., R.H. and M.J.M. conceived the project. M.B., M.W., D.D., T.H., L.T., M.J.M. and R.H. performed the experiments. All authors analysed the results. M.B., T.H., M.J.M. and R.H. wrote the paper.

## Competing interests

The authors M.W. and M.J.M. are inventors on patent application WO 2020074915 for the iCaspase9 transgenic chicken. The University of Edinburgh is the applicant. The remaining authors declare no competing interests.
