## [Peer Review File · Nature Communications]

REVIEWER COMMENTS

Reviewer #1 (Remarks to the Author):

This is an interesting paper that describes a useful approach to efficiently transfer existing and introduce novel beneficial alleles and generate homozygous (or heterozygous) targeted gene edits in poultry using Sire Dam Surrogate (SDS) mating. I have only minor comments. Please see attached review document

Regards

Alison Van Eenennaam

Reviewer #2 (Remarks to the Author):

The authors have demonstrated a novel surrogate system so called Sire Dam Surrogate (SDS) mating based on inducible germ cell sterile surrogate chicken using iCaspase9. The authors successfully integrated iCaspase9 and aviCaspase9 gene cassette into chicken germ cell marker gene, DAZL, which could induce germ cell specific cell death. Using the transgenic chicken lines, they successfully recreates pure chicken breeds in terms of appearance and molecular characteristics. Although the author demonstrated the successful reconstitution of chicken breeds using the transgenic chicken lines, however, this paper has the limitation in two major aspects. First of all, I can't agree with that the SDS breeding greatly accelerates the production of homozygote offspring as the authors mentioned. As their claim, homozygous offspring from genome edited chickens from first-generation heterozygous chickens may take 5-6 months, but considering the periods of germ cell isolation, maintenance, in ovo incubation using surrogate eggs and the production efficiency, there is no significant difference in time and cost. Rather, it cannot be said to be an efficient method if the cost of introducing or using the system is considered. Furthermore, for the conservation of existing chicken breeds, it is much easier to produce them through natural breeding.

On the other hand, the significance of this paper can be found in the aspect of diversity of bioengineering techniques, but it cannot be seen as a particularly efficient method in terms of progressiveness. Since the results could not provide sufficient striking advance compared to the cDDX4 knockout chicken as surrogate system that is previously reported from same research group, which has a similar purpose (Woodcock et al. 2019, PNAS. Reviving rare chicken breeds using genetically engineered sterility in surrogate host birds), this paper only adopts two methods including CRISPR/Cas9 and inducible gene targeting, but it is difficult to regard it as a scientific advance. Collectively, this paper is considered to be insufficient in terms of scientific progress and practical proposals.

Specific Comments

1. What is the benefit of the iCaspase9 and aviCaspase9 transgenic chickens compared to cDDX4 knockout chickens which is previously reported (Woodcock et al., 2019, PNAS)? The efficiency looks similar.
2. Only 22% of the G1 offspring (8/74) contained the caspase9 transgene? If the donor PGC were injected into cDDX4 knockout chicken line, it should be around 50%. The result suggests there is remaining oocytes in cDDX4 knockout female chicken? Please clarify it.
3. The transgenic chickens used in this study are heterozygous or homozygous? The expression level of caspase9 may affect to the efficiency of germ cell death.
4. Several conditional knockout system reported leaky effects without induction. There is any germ cell death, or cell death in the other tissues of the iCaspase and aviCaspase transgenic chickens? Please provide the number of germ cells and the number of cells in several organs.
5. Please revise Supplementary Table 1. There are many typos and missing words.
6. What does “correct genotype host” means in Supplementary Table 1? The number includes donor PGC-derived progenies?
7. Supplementary Table 2 showed that transgenic chickens were hatched from only DDX4 knockout female chicken, not in ZW wild type recipient. What does it mean? Please explain it.

Reviewer #3 (Remarks to the Author):

The authors present a well written manuscript describing an interesting topic relevant to a broad readership. By producing chickens with an inducible depletion of germ cells they generate for the first time a sterile male host for germ cell transplantation. In combination with the previously

published sterile DDX4^{-/-} hens it is now possible, as shown in the manuscript, to produce homozygous edited chickens straight from chimeric male and female chickens with a high efficiencies. This has on the one hand side the advantage of speeding up the process of generating genetically modified chickens, increasing the efficiency and on the other hand side gives the opportunity to restore endangered or extinct chicken lines. The described results are a major step towards conservation of rare bird species.

Minor comments:

- 1) after injection of edited PGCs and B/B the authors see hatching rates slightly above 50%. Have the authors tried to inject B/B alone into edited as well as wild type embryos to analyze the effect of B/B alone as well as determine the DMSO concentration that is tolerated by the embryos?
- 2) Fig 1c. The gating strategy is not clear. It says SSEA1 expression but in the figure legend it is stated that co-expression of GFP + SSEA1 was analyzed. Please show a dot plot or the complete gating strategy.
- 3) For the first time expression under a germline specific promoter (cDAZL) was used in chickens. The authors state that expression was seen in germ cells and the developing gonads. Have other time points and tissues being examined?
- 4) Are homozygous iCaspase9 chickens developing comparable to wild type siblings and is there any leakiness of the system seen without induction by B/B?

We thank the reviewers for appreciating the importance of the research and the many suggestions to improve the manuscript. We have now been allowed into our laboratory and have carried out a guide off-target analysis and added this data in Supplementary Figures 10 and 13. We have also changed the title to make the manuscript more accessible to the public and added some minor edits to the main text to improve overall clarity. Several new supplementary figures were added to address the concerns of the reviewers. We describe our response to each reviewer below.

Reviewer 1:

We thank this reviewer for taking the time for the thorough reading of the manuscript to find the large number of errors. This reviewer only had minor changes. We detail our changes below.

1. *I am not sure about the sentence at the end of the abstract ... "reconstitution of chicken breeds carrying desired genetic changes" sounds like the breeds are somehow "reconstituted", rather than being produced from the edited PGCs? Perhaps different wording would make them sound less like dehydrated chickens that just need water added to be reconstituted....*

We have rewritten this sentence and we now use the term 'restore' to signify 're-establishing' pure breeds.

2. *Lack of consistency in naming of genes throughout manuscript e.g. pmel17, PMEL17, pMel17 – and consistent for all gene names;*

We have gone through the manuscript and corrected chicken gene and protein names and nomenclature.

3. *Be consistent with description of this in terms of capital letters iCaspase9 and aviCaspase9 Define your abbreviations and use them consistently e.g. LSX, FRZ, DOW –*

We have made these changes.

4. *Page 2. End of third paragraph "...for an introduced genetic variation" should be genetic variant*

This was corrected.

5. *Page 6 Normally list animal welfare protocol number specifically associated with this project –*

We have added our project licence number to the Methods.

6. *Page 7 line 2 Whyte et al. ?? not linked to reference list – which paper? –*

This was added to the reference list.

7. *Clinton et al. ?? not linked or present in reference list*

This was added to the reference list.

8. **Table 1.** *Was unclear what Day 10 embryo numbers represent? Are they a subset of eggs set? Need to add “&Eggs examined at day 10 of incubation for fertility” Use the term fertile for the †Fertile eggs with embryos present at day 18 of incubation so that it does not get confused with “fertility” which is what was done with the day 10 embryos.*

We have now removed the embryo data in which we observed eggs at day 10 of incubation for fertility as it complicated the table. We now only show the fertility data from eggs taken to hatch and for which the fertility is measured at day 18 of incubation.

9. *Need to have capital iCaspase in the column one headings on Table 1 and 2*

We have corrected this.

10. **Figure 1. b** – *Is there a significant difference in cell count per well for the difference treatments at 0 nM B/B? why?*

We have added additional data and now carried out a statistical analysis. The controls show no statistical difference for B/B = 0 and iCaspase9 cells show differences from control cells for all treatment doses.

11. **Figure 1c** *What is the Y axis? Unclear what this figure legend means “10day gonads were examined for co-expression of GFP and SSEA-1; blue, no primary control.” What does no primary control mean? What is 97.7% meant to represent? –*

We have replaced this figure with a version showing the axis. The supplementary Figure 3 now shows the windows and the population of cells in each window.

12. **Figure 1d** *what does the Scale bar represent? What was the wavelength used to excite? There is no description of this work in materials and methods or the Supplementary Materials and Methods. Typically I think the type of microscope is also meant to be listed in Nature publications. –*

We have added microscope and antibody data to the Mats & Methods.

13. **Figure 2** *b-d should be b,d; c-e should be c,e –*

This is now corrected.

14. **Figure 3** *Very hard to read and decipher this figure as all of the dots are all over each other and most of the image is white space. Think of a better way to present this data. Perhaps blow up the clusters so can see all the dots in a cluster. There is no colour that looks grey. If you are going to use abbreviations like LSX and DOW in the legend then use it consistently. The labels on the graph were hard to read and too small. Perhaps use squares to indicate host breeds. Triangles to indicate controls. And circles to indicate breeds that were used as PGCs. –*

We have used the reviewer’s useful suggestions to improve the figure. The data point labels were enlarged and changed to triangles, squares, and circles. We think the figure is greatly improved.

15. **Figure 4** *again define abbreviations in the legend and use them consistently....LSX, FRZ*

There are three times XXXbp is written in the legend – assume these all need to have a value? –

These mistakes have been corrected.

16. **Figure 5** *Don't the iCaspase9 host embryos need to be treated with B/B in the dorsal aorta of stage 16 chicken embryos? that should be represented in the schematic? –*

The reviewer is correct. We have added this information to the figure.

17. **Supplementary Figure 1c** *why are the PCR bands for the G1 offspring a different size to those of the PGCs? What is the expected size of that band? Is the MW marker lined up appropriately for the right side of the gel? –*

We have now added the expected PCR products and added size markers to the gel. There was a slight skew to the gel (shown below) but the space in the figure does indicate that the central lanes of the gel have been removed.

18. **Supplementary Figure 2** *iCaspas9 spelled incorrectly –*

This was corrected.

Reviewer 2 does not believe Surrogate sire and surrogate dam mating accelerates the generation of homozygote genome edited offspring due to the time needed for culturing the PGCs during genome editing nor that this method reduces overall costs.

We beg to differ on both of these points. Housing of research chicken and livestock is extremely expensive in comparison to mice or other research models. Any methods that reduce the time needed to maintain these animals or reduce animal numbers greatly reduces costs and addresses the 3Rs simultaneously. The time taken to clonally target the PGCs (six weeks) is a laboratory research expense and is vastly cheaper than livestock housing expenses. Screening for the desired edit in PGCs before producing offspring also reduces animal numbers used in research. Validated genome edited chicken can be cryopreserved and stored until animal funding is obtained to turn the edited PGCs into edited chicken.

Furthermore, homology directed repair is currently not possible when injecting genome editing tools into the early chicken embryo.

As an example, the current alternative technology for genome editing birds is injection of adenovirus carrying CRISPR/Cas9 into the blastoderm of laid G₀ eggs. This was shown elegantly by Lee et al, PNAS, 2019. This method generates independent INDELS in the G₁ birds. The G₁ birds can be selected and mated to generate homozygote G₂ birds containing a knockout of the gene of interest. In the case above, the authors were lucky to produce two independent G₁ birds that were male and female and contained an identical deletion. If not, the G₁ bird would need to be mated to a wildtype bird to generate G₂ birds containing the same heterozygote deletion. These 2 birds would be mated to generate G₃ birds, 25% containing the homozygote deletions. As these experiments were done in quail, the generation time is quicker but in chicken it would take 12 to 18 months to generate homozygote offspring with identical random deletions. We believe our PGC system is highly advantageous over blastodermal injections.

The second point is that conserving existing chicken breeds is easier to do through natural breeding. I point to the seminal opinion paper by Mary Delany published in 2003 in Science (Vol. 300, pp. 1667-1668) that research chicken lines are in a desperate state and are being lost due to lack of funding and because the technology for cryopreservation of poultry is woefully lacking. 17 years have passed since this paper and this situation has not changed or improved. Natural breeding will not help for the long term preservation of traditional, indigenous, and research lines of chicken. We hope that our surrogate hosts will offer an alternative. We have added some of these points to the discussion section.

Specific

Comments

1. What is the benefit of the iCaspase9 and aviCaspase9 transgenic chickens compared to cDDX4 knockout chickens which is previously reported (Woodcock et al., 2019, PNAS)? The efficiency looks similar.

The main difference is the previous DDX4 line cannot be used to produce a sterile male as DDX4 is on the Z sex chromosome. The iCaspase9 locus is on chromosome 3 so both sterile male and female chicken can be bred. The eggs laid per week by the DDX4 surrogate in these experiments is 3.33 eggs per week and is shown in Supplementary Table 2. The eggs laid per week for the iCaspase9 surrogates was 5.13-7.0 eggs per week. This is probably due to the DDX4 ZW endogenous germ cells, which are lost post hatch thus can compete with the donor germ cells until that point. The iCaspase9 germ cells are ablated at 2.5 days of incubation which gives donor germ cells increased time to populate the gonad before undergoing meiosis at day 14 of incubation.

50% of the female offspring bred from the DDX4 ZZ male contain no endogenous germ cells (DDX4 ZW) and 50% are wildtype. While using a homozygote iCaspase9 male for mating, 100% of the female (and male) fertile eggs are iCaspase9 positive (heterozygous iCaspase9) and can be used as sterile surrogate hosts. This information is now shown in the Supplementary Table 1 and in the discussion. Furthermore, with both an iCaspase9 male and female surrogate hosts, they can be directly mated to produce homozygous genome edited offspring.

2. Only 22% of the G1 offspring (8/74) contained the caspase9 transgene? If the donor PGC were injected into cDDX4 knockout chicken line, it should be around 50%. The

result suggests there is remaining oocytes in cDDX4 knockout female chicken? Please clarify it.

We expected 50% of the offspring to derive from the donor germ cells. The postdoc informs me that the PGCs were not gated correctly and a percentage of non-GFP PGCs were in the sorted population. These non-GFP cells must have out competed the GFP⁺ cells. We have changed the results section on page 3 to reflect that some sorted cells were not GFP⁺.

3. The transgenic chickens used in this study are heterozygous or homozygous? The expression level of caspase9 may affect to the efficiency of germ cell death.

We agree with this statement. We only used and tested heterozygote iCaspase9 embryos as surrogate hosts and for ablation experiments. We have made this clear in the Materials and Methods on page 7.

4. Several conditional knockout system reported leaky effects without induction. There is any germ cell death, or cell death in the other tissues of the iCaspase and aviCaspase transgenic chickens? Please provide the number of germ cells and the number of cells in several organs.

We have carried out a Western blot analysis on different tissues from day 10 embryos (Supplementary Fig. 4) and only detected GFP expression in the gonad. We used an anti-cleaved caspase3 antibody on B/B treated and control embryos 7 days post treatment and did not detect additional caspase3 positive cells in the gonad (Supplementary Fig. 8). We measured the lay rate of iCaspase9 heterozygote hens and did not see a reduction in lay rate in comparison to control hens (Supplementary Table 3) indicating that the germ cells/oocytes are developing normally in the iCaspase9 chicken.

5. Please revise Supplementary Table 1. There are many typos and missing words.

We have revised all tables in accordance with reviewer 3 below.

6. What does "correct genotype host" means in Supplementary Table 1? The number includes donor PGC-derived progenies?

We agree that this title is confusing. We have changed this to 'Hatched host chicks with a sterile genotype' and extended the description of the genotypes generated from mating the transgenic sire males.

7. Supplementary Table 2 showed that transgenic chickens were hatched from only DDX4 knockout female chicken, not in ZW wild type recipient. What does it mean? Please explain it.

Similar to mouse ES cells, PGCs lose the competence for germ line transmission the longer they remain in culture. In this case, the iCaspase9 integrated PGCs could not compete with the endogenous PGCs of the wildtype host. We observed the same result in our Woodcock PNAS article (2019) using donor heritage broiler PGCs. This was one of the primary reasons that we first decided to generate sterile host surrogate birds for donor PGCs. We have added text on page 3 to explain this result.

Reviewer 3:

We thank this reviewer for seeing the benefits of our system for the ‘speeding up the process of generation genetically modified chicken, increasing the efficiency...’ and giving ‘the opportunity to restore endangered or extinct chicken lines’ and ‘a major step towards conservation of rare bird species.’

This reviewer only had minor comments:

1) after injection of edited PGCs and B/B the authors see hatching rates slightly above 50%. Have the authors tried to inject B/B alone into edited as well as wild type embryos to analyze the effect of B/B alone as well as determine the DMSO concentration that is tolerated by the embryos?

Our poor hatch rate is most likely from hatching windowed eggs. We have varied the incubation rocking conditions and our hatch rates have now improved. We intend to publish a techniques paper describing the windowing and incubation process in detail.

The B/B compound is known to be innocuous in many cell and animal models. However, it is recommended by the manufacturer to be dissolved in ethanol. Stage 16 HH embryos are robust and we can inject 1ul of 50% ethanol into the bloodstream without adverse effects. Likewise, we have noted no adverse effects of the B/B compound on chicken embryos or cells. However, 1 ul of 100% ethanol added to the 50 ul of donor PGC suspension (2% f.c.) kills the donor PGCs. We have found that B/B compound is also soluble in DMSO in place of ethanol. We use a 5% DMSO solution for freezing PGCs. We added 1 ul of DMSO to 50 ul of GFP⁺ PGC suspension (2% f.c.) and let the solution sit at RT for one hour. The cell suspension was added back to a cell culture well and observed the next day. The PGC survival (by observing GFP⁺ cells) appeared equal to control cells. We have added the information that ethanol is toxic to PGCs to the Materials and Methods.

2) Fig 1c. The gating strategy is not clear. It says SSEA1 expression but in the figure legend it is stated that co-expression of GFP + SSEA1 was analyzed. Please show a dot plot or the complete gating strategy.

We have replaced Fig. 1c with a panel showing GFP expression and SSEA1 immunofluorescence. The gated populations are quantified in Supplementary Fig. 3.

3) For the first time expression under a germline specific promoter (cDAZL) was used in chickens. The authors state that expression was seen in germ cells and the developing gonads. Have other time points and tissues being examined?

We examined post hatch gonads and embryos. The sections of adult iCaspase9 testes showed only a low GFP fluorescence. We have added a Western blot (mentioned above) using tissues from a day 10 of incubation embryo.

4) Are homozygous iCaspase9 chickens developing comparable to wild type siblings and is there any leakiness of the system seen without induction by B/B?

We have observed no health or welfare issues or unexplained deaths in the iCaspase9 chicken. We monitor chick growth for the first two weeks of life for new GE lines and have observed no slow growth or lethargy in the iCaspase9 chicks. We have kept the iCaspase9 female hosts injected with B/B compound plus donor *PMEL17* edited germ cells for two years and observed no detrimental health effects. We routinely use male iCaspase9 homozygotes mated to wildtype female for producing fertile heterozygote eggs for injection. We obtain good fertility from these males. We have added a supplementary Table 3 demonstrating that the egg laying rate for the heterozygote iCaspase9 hens is equivalent to wildtype hens. In the future, when we re-generate the line it would be useful to monitor the lay rate in homozygote female hens to determine if oocyte development is in any way affected.

REVIEWERS' COMMENTS

Reviewer #2 (Remarks to the Author):

Previous concerns have been addressed properly